# MODiR: Multi-Objective Dimensionality Reduction for Joint Data Visualisation

## Abstract

Many large text collections exhibit graph structures, either inherent to the content itself or encoded in the metadata of the individual documents. Example graphs extracted from document collections are co-author networks, citation networks, or named-entity-cooccurrence networks. Furthermore, social networks can be extracted from email corpora, tweets, or social media. When it comes to visualising these large corpora, either the textual content or the network graph are used.

In this paper, we propose to incorporate both, text and graph, to not only visualise the semantic information encoded in the documents' content but also the relationships expressed by the inherent network structure. To this end, we introduce a novel algorithm based on multi-objective optimisation to jointly position embedded documents and graph nodes in a two-dimensional landscape. We illustrate the effectiveness of our approach with real-world datasets and show that we can capture the semantics of large document collections better than other visualisations based on either the content or the network information.

## 1 Introduction

Substantial amounts of data is produced in our modern information society each day. A large portion of it comes from the communication on social media platforms, within chat applications, or via emails. This data exhibits dualtiy in the sense that they can be represented as *text* and *graph*. The metadata provides an inherent graph structure given by the social network between correspondents and the exchanged messages constitute the textual content. In addition, there are many other datasets that exhibit these two facets. Some of them are found in bibliometrics, for example in collections of research publications as co-author and citation networks.

When it comes to analyse these types of datasets, usually either the content or the graph structure is neglected. In data exploration scenarios the goal of getting an overview of the datasets at hand is insurmountable with current tools. The sheer amount of data prohibits simple visualisations of networks or meaningful keyword-driven summaries of the textual content. Data-driven journalism (Coddington, 2015) often has to deal with leaked, unstructured, very heterogeneous data, e.g. in the context of the Panama Papers, where journalists needed to untangle and order huge amounts of information, search entities, and visualise found patterns (Chabin, 2017). Similar datasets are of interest in the context of computational forensics (Franke & Srihari, 2007). Auditing firms and law enforcement need to sift through huge amounts of data to gather evidence of criminal activity, often involving communication networks and documents (Karthik et al., 2008). Users investigating such data want to be able to quickly gain an overview of its entirety, since the large amount of heterogeneous data renders experts' investigations by hand infeasible. Computer-aided exploration tools can support their work to identify irregularities, inappropriate content, or suspicious patterns. Current tools[1] lack sufficient semantic support, for example by incorporating document embeddings (Mikolov et al., 2013) and the ability to combine text and network information intuitively.

We propose *MODiR*, a scalable multi-objective dimensionality reduction algorithm, and show how it can be used to generate an overview of entire text datasets with inherent network information in a single interactive visualisation. Special graph databases enable the efficient storage of large relationship networks and provide interfaces to query or analyse the data. However, without prior

---

[1]e.g. `https://www.nuix.com/` or `https://linkurio.us/`

knowledge, it is practically impossible to gain an overview or quick insights into global network structures. Although traditional node-link visualisations of a graph can provide this overview, all semantic information from associated textual content is lost completely.

Technically, our goal is to combine network layouts with dimensionality reduction of high-dimensional semantic embedding spaces. Giving an overview over latent structures and topics in one visualisation may significantly improve the exploration of a corpus by users unfamiliar with the domain and terminology. This means, we have to integrate multiple aspects of the data, especially graph and text, into a single visualisation. The challenge is to provide an intuitive, two-dimensional representation of both the graph and the text, while balancing potentially contradicting objectives of these representations.

In contrast to existing dimensionality reduction methods, such as tSNE (Maaten & Hinton, 2008), *MODiR* uses a novel approach to transform high-dimensional data into two dimensions while *optimising multiple constraints* simultaneously to ensure an optimal layout of semantic information extracted from text and the associated network. To minimise the computational complexity that would come from a naive combination of network drawing and dimensionality reduction algorithms, we formally use the notion of a hypergraph. In this way, we are able to move repeated expensive computations from the iterative document-centred optimisation to a preprocessing step that constructs the hypergraph. We use real-world datasets from different domains to demonstrate the effectiveness and flexibility of our approach. *MODiR*-generated representations are compared to a series of baselines and state-of-the-art dimensionality reduction methods. We further show that our integrated view of these datasets exhibiting duality is superior to approaches focusing on text-only or network-only information when computing the visualisation.

## 2 RELATED WORK

With *MODiR* we bridge the gap between text and network visualisation by jointly reducing the dimensionality of the input data. Therefore we subdivided this part into three sections to highlight related work in the areas of text visualisation, representation learning, as well as dimensionality reduction. Other work that tries to jointly model text and networks but without dimensionality reduction and without a focus on visualisation is *LINE* (Tang et al., 2015). They generate information networks consisting of different types of nodes, e.g. words from document content and authors from document metadata. Another tool that investigates combining graph structure with textual elements is *VOSviewer* (Van Eck & Waltman, 2014). They construct and visualise bibliographic networks that provide a multi-view interface to explore and filter keywords and network aspects of such datasets. In our work we go beyond building a network from textual data but instead project the textual data into a latent space.

Document visualisation aims to visualise the textual content, such that users gain quick insights into topics, latent phrases, or trends. Tiara (Wei et al., 2010) extracts topics and derives time-sensitive keywords to depict evolving subjects over time as stacked plots. Another line of work projects documents into a latent space, for example by using topic models or embeddings: Creating scatter-plots of embedded documents of a large corpus may result in a very dense and unclear layout, so Chen et al. (Chen et al., 2009) developed an algorithm to reduce over-full visualisations by picking representative documents. A different approach is taken by Fortuna et al. (Fortuna et al., 2005), who do not show documents directly, but generate a heatmap of the populated canvas and overlay it with salient phrases at more densely populated areas from the underlying documents in that region. Friedl et al. (Fried & Kobourov, 2014) extend that concept by drawing clear lines between regions and colouring them. They also add edges between salient phrases based on co-occurrences in the texts. A map analogy can be used to visualise the contents of documents by embedding them into a high dimensional semantic space (Le & Mikolov, 2014) and projecting it on a two-dimensional canvas as a *document landscape*. Most recently *Cartograph* (Sen et al., 2017) was proposed, which is visually very similar to previous approaches, but pre-renders information at different resolution and uses a tiling server with (geographic) map technology to deliver responsive interactions with the document landscape. Regions are coloured based on underlying ontologies from a knowledge-base. Networks are traditionally visualised using so-called node-link graphs. This way, any additional information related to nodes and edges are lost. The layout of nodes usually follows a force-based analogy first proposed by Fruchterman & Reingold (1991). Newer approaches optimise the compu-

tational complexity and include local metrics to better represent inherent structures as for example *ForceAtlas2* (Jacomy et al., 2014), which is the default network algorithm for the network visualisation tool *Gephi*.

The text and network visualisation methods discussed above primarily use structural properties of the data to generate their layout. Although we focus on the visualisation of text data with inherent graph information, *MODiR* can work with arbitrary kinds of data. Our model only requires a way to project the data into a high-dimensional Euclidean vector space so that the distance between two points can be interpreted as their (semantic) similarity. Traditionally, text can be represented as bag-of-words vector that optionally is weighted by respective tf-idf scores. In recent years, embeddings became more popular as they conserve semantic meaning in their vector representation. Mikolov et al. (2013) introduced neural architectures to learn high-dimensional vector representations for words and paragraphs (Le & Mikolov, 2014). Similar methods are used to learn representations for nodes in a network based on either the structural neighbourhood (Faerman et al., 2018) or additional heterogeneous information (Chang et al., 2015).Schlötterer et al. (2017) attempted to learn joint representations of network structure and document contents but saw no improvement over conventional models in a series of classification tasks. We only use the structural information of the network for better control over fine-grained adjustments in our layout algorithm.

The goal of dimensionality reduction is to represent high-dimensional data in a low-dimensional space while preserving the characteristics of the original data as sound as possible. A very common application of dimensionality reduction is to project high-dimensional data into two dimensions for the purpose of visual interpretation. Generally, these methods follow one of three mathematical models. *Linear* models, such as Principle Component Analysis (PCA) (Pearson, 1901) can be calculated very efficiently and have proven to reduce input spaces to improve the performance of downstream tasks. Thus, they are often indirectly used for feature extraction. Although reductions to two dimensions for visualisations are appropriate for quick initial data exploration, other approaches are able to better preserve data characteristics in two dimensions. For example, the *non-linear* Sammon mapping (Sammon, 1969) tries to preserve the structure of inter-point distances in high-dimensional space in low-dimensional space. The resulting visualisations are generally better then PCA to show relatedness of individual data points. Lastly, there are *probabilistic* models like Stochastic Neighbour Embeddings (SNE) (Hinton & Roweis, 2003). They are similar to a Sammon mapping in that they use inter-point distances but model these distances as probability distributions. The t-distributed SNE has proven to produce competitive results for visualising datasets while preserving characteristics (Maaten & Hinton, 2008), however its nondeterministic nature may produce greatly varying results. Recently, *FltSNE* was proposed, an optimisation of tSNE that significantly reduces the computational complexity (Linderman et al., 2019). Other newer dimensionality reduction algorithms like *LargeVis* (Tang et al., 2016) and *UMAP* (McInnes et al., 2018) scale almost linearly by using efficient nearest neighbourhood approximations in the high-dimensional space and spectral embeddings to initialise positions of points in the low-dimensional space to reduce the number of fine-tuning iterations.

## 3 MULTI-OBJECTIVE DIMENSIONALITY REDUCTION

Visualisations of complex datasets are restricted to two or three dimensions for users to grasp the structure and patterns of the data. We integrate multiple entities (i.e., documents and persons) into a joint visualisation, which we call *landscape*. This landscape consists of a base-layer containing all documents depicted as dots forming the *document landscape*; nodes and their connections are placed on top of this base-layer as circles connected by lines forming the *graph layer*. In this section, we propose the *MODiR* algorithm which integrates multiple objectives during the layout process to find an overall good fit of the data within the different layers. Our approach is derived from state-of-the-art methods for drawing either the network layer or the document landscape.

We assume that documents are given as high-dimensional vectors and entities are linked among one another and to the documents. These links are used as restrictions during the multi-objective dimensionality reduction of document vectors. Let $\boldsymbol{x}^{(i)} \in \mathbb{X} \subset \mathbb{R}^d$ be the set of $n$ documents in their $d$-dimensional representation and $\boldsymbol{y}^{(i)} \in \mathbb{Y} \subset \mathbb{R}^2$ the respective positions on the document landscape. Let $\mathcal{H}(\mathcal{V}, \mathcal{E})$ be a hypergraph based on the network information inferred from the document corpus, with vertices $\mathcal{V} = \mathbb{X} \bigcup \mathbb{P}$, where $\mathbb{X}$ are the documents and $p_i \in \mathbb{P}$ are the entities in the

network and hyperedges $e_k \in \mathcal{E}$ describing the relation between documents and entities. For each pair of entities $p_m, p_n \in \mathbb{P}$ that are connected in the context of documents $\boldsymbol{x}^{(i)}, \ldots \in \mathbb{X}$, there is a hyperedge $e_k = \{p_m, p_n, \boldsymbol{x}^{(i)}, \ldots\}$. Analogously, the same definition applies to $\mathbb{Y}$. Further, $\mathcal{H}^{\mathbb{Y}}$ or $\mathcal{H}^{\mathbb{X}}$ is used to explicitly state the respective document representation used. The position in the graph layer $\pi : \mathbb{P} \to \mathbb{R}^2$ of an entity $p_m$ is defined as

$$\pi(p_m; \mathcal{H}^{\mathbb{Y}}) = \frac{1}{N_{p_m}} \sum_{e_k \in \mathcal{E}_{p_m}} \sum_{\boldsymbol{y}^{(i)} \in e_k \setminus \mathbb{P}} \boldsymbol{y}^{(i)}, \tag{1}$$

where $\mathcal{E}_{p_m} \subset \mathcal{H}^{\mathbb{Y}}$ is the set of hyperedges containing $p_m$ and $N_{p_m}$ is the number of documents $p_m$ is associated with.[2] This effectively places an entity at the centre of its respective documents. More elaborate methods like a density-based weighted average are also applicable to mitigate the influence of outliers. For simplicity we will abbreviate $\pi(p_m; \mathcal{H}^{\mathbb{Y}})$ as $\pi_m$.

Let $\psi : \mathbb{X} \to \mathbb{Y}$ be the projection $\psi(\boldsymbol{x}^{(i)}; \boldsymbol{W}) = \boldsymbol{W}_{i,:} = \boldsymbol{y}^{(i)}$, where $\boldsymbol{W} \in \mathbb{R}^{2 \times n}$ is the projection matrix leant by *MODiR* based on multiple objectives $\varphi_{\{1,2,3\}}$ using gradient descend, as defined later in this section. The objectives are weighted by manually set parameters $\theta_{\{1,2,3\}}$ to balance the effects that favour principles focused on either the graph layer or the document landscape, as they may contradict one another. Given a high-dimensional hypergraph $\mathcal{H}^{\mathbb{X}}$, the matrix $\boldsymbol{W}$, and a entity projection $\pi$, we define the resulting multi-objective dimensionality reduction function as

$$\Psi(\mathcal{H}^{\mathbb{X}}, \boldsymbol{W}, \pi) = \mathcal{H}^{\mathbb{Y}}.$$

In the following paragraphs, we will formally introduce *MODiR*'s objectives. *Objectives (1) and (2) are inspired by tSNE and use the neighbourhood context of documents in $\mathbb{X}$ to position similar documents near one another and unrelated ones further apart in $\mathbb{Y}$. Objective (3)* attracts documents based on co-occurrence in hyperedges so that the resulting $\pi_m$ will be closer if they are well connected in the graph. This third objective also implicitly brings documents closer to their respective entities.

**Objective (1): Similar documents are near one another.**    Semantically similar documents should be closer on the document landscape and dissimilar ones further apart. To measure the semantic similarity of documents, Maaten & Hinton (2008) used a naïve bag-of-words representation. Although tSNE preserves the inherent semantic structure in two-dimensional representations from these sparse vectors (Pezzotti et al., 2017), we opted to use document embeddings. This has the advantage that, when only part of the data is visualised, the embedding model can still be trained on a larger set of documents and thus retain the additional information. Objective (1) is inspired by the efficient usage of context words in word2vec (Mikolov et al., 2013). Corresponding to the skip-gram model, we define the context $\mathbb{X}^{k, \boldsymbol{x}^{(i)}} \subset \mathbb{X}$ of a document $\boldsymbol{x}^{(i)}$ by its $k$ nearest neighbours in the embedding space. The first objective is defined as

$$\varphi_1(x^{(i)}) = \sigma\Big( \sum_{\boldsymbol{x}^{(j)} \in \mathbb{X}^{k, \boldsymbol{x}^{(i)}}} \|\boldsymbol{x}^{(i)} - \boldsymbol{x}^{(j)}\| - \|\boldsymbol{y}^{(i)} - \boldsymbol{y}^{(j)}\| \Big) \tag{2}$$

with $\sigma$ being the sigmoid function. Distances are normalised based on the context to make them comparable between the high-dimensional and two-dimensional space and rescaled by the sigmoid.

**Objective (2): Dissimilar documents are apart from one another.**    The optimal solution to the previously defined objective would be to project all documents onto the same point on the two-dimensional canvas. In order to counteract that, we introduce negative examples for each pair of context documents. We do so by sampling a set of $l$ documents that are not in the $k$ neighbourhood of $\boldsymbol{x}^{(i)}$. Let $\bar{\mathbb{X}}^{l, \boldsymbol{x}^{(i)}} \subset \mathbb{X} \setminus \mathbb{X}^{k, \boldsymbol{x}^{(i)}}$ be the set of negative samples for $\boldsymbol{x}^{(i)}$, then the second objective is defined as

$$\varphi_2(\boldsymbol{x}^{(i)}) = \sigma\Big( \sum_{\boldsymbol{x}^{(j)} \in \bar{\mathbb{X}}^{l, \boldsymbol{x}^{(i)}}} \|\boldsymbol{x}^{(i)} - \boldsymbol{x}^{(j)}\| - \|\boldsymbol{y}^{(i)} - \boldsymbol{y}^{(j)}\| \Big). \tag{3}$$

---

[2] $N_{p_m} := \Big| \{\boldsymbol{x}^{(i)} \in \mathbb{X} | \exists e_k \in \mathcal{E} : \boldsymbol{x}^{(i)} \in e_k \wedge p_m \in e_k\} \Big|$

**Objective (3): Connected entities are near one another and their documents.** This object serves two purposes: All documents $\boldsymbol{y}^{(i)}$ associated with a person $p_m$ are placed near its $\pi_m$ position in the graph layer and two people $\pi_m$ and $\pi_n$ are forced near one another if they are connected.

Let $\mathcal{E}_{\boldsymbol{y}^{(i)}} \subset \mathcal{E}$ be the set of hyperedges in the hypergraph $\mathcal{H}$ containing the document $\boldsymbol{y}^{(i)}$ and $\mathcal{E}_{\boldsymbol{y}^{(i)}}^{\mathbb{Y}} = \bigcup_{e_k \in \mathcal{E}_{\boldsymbol{y}^{(i)}}} e_k \setminus \mathbb{P}$ all documents that are linked to $\boldsymbol{y}^{(i)}$ through an entity, then the third objective is defined as

$$\varphi_3(\boldsymbol{y}^{(i)}) = \sigma\Big( \sum_{\boldsymbol{y}^{(j)} \in \mathcal{E}_{\boldsymbol{y}^{(i)}}^{\mathbb{Y}}} \|\boldsymbol{y}^{(i)} - \boldsymbol{y}^{(j)}\| \Big), \tag{4}$$

which, when minimised, attracts documents that are related through entities. This has two implicit effects: An entity $p_m$ gets closer to its documents as they are attracted to $\pi_m$ without having to explicitly compute this position using Equation 1. Also, related entities $p_m, p_n$ are attracted to one another since they appear in the same hyperedges. The computational complexity of this objective is strongly related to the connectedness of entities in the graph. For dense graphs, we propose a heuristic by only using a subset of $s$ documents from the context $\mathcal{E}_{\boldsymbol{y}^{(i)}}^{\mathbb{Y}}$ of $\boldsymbol{y}^{(i)}$. An objective modelling a repulsive force as in force-directed graph layouts is not needed as the first two objectives $\varphi_{\{1,2\}}$ provide enough counteracting force.

**Algorithm.** The positions of entities and documents on the landscape are calculated using the previously defined objectives as follows. First, we construct the hypergraph $\mathcal{H}^{\mathbb{X}}$ with document contexts including the set of $k$-neighbourhoods $\mathbb{X}^{k,\boldsymbol{x}^{(i)}}$. Relevant pairwise distances can be stored in an adjacency matrix so reduce computational overhead in Equations 2 and 3. For more efficient training, the randomly sampled $l$ negative neighbourhoods $\bar{\mathbb{X}}^{l,\boldsymbol{x}^{(i)}}$ can be prepared ahead of time and then only masked during later. The $s$-neighbourhoods for entities in Equation 4 $\mathcal{E}_{\boldsymbol{y}^{(i)}}^{\mathbb{Y}}$ can only be prepared with references, as $\mathbb{Y}_{\boldsymbol{y}^{(i)}}$ updates with each iteration. We designed the algorithm to move as much repetitive computations to pre-processing ahead of time or each epoch. Creating these sets is very efficient using Hierarchical Navigable Small World graphs (HNSW) (Baranchuk et al., 2018) for approximate nearest neighbour search. Overall we are able to reduce the pre-processing complexity to $\mathcal{O}(n \log n)$ and for each iteration $\mathcal{O}(kln)$, with $k, l \ll n$ near linear. After generating the context sets, we use gradient descend to update the projection matrix $\boldsymbol{W}$ with learning rate $\eta$ reducing the overall error $\Phi$ as defined by

$$\Phi(x_i) = \theta_1 \varphi_1(x_i) + \theta_2 \varphi_2(x_i) + \theta_3 \varphi_3(x_i). \tag{5}$$

Selecting appropriate values for the hyperparameters $k$, $l$, $s$, and $\theta_{\{1,2,3\}}$ is critical to produce meaningful results. We found $l = k$ in all experiments to produce the best results as this way for every similar document the model has one dissimilar document to compare. Inspired by tSNE (Maaten & Hinton, 2008), we limit hyperparameters by setting $k$ and $s$ dynamically for each document based on a user-defined perplexity. With these adaptations, the only parameters to be set are the perplexity $\beta$ that roughly determines the context size, the learning rate $\eta$, and the objective weights, which can often stay at a default setting. A reference implementation including a modular processing pipeline for different datasets, approaches, and experiments is available on GitHub[3].

# 4 EXPERIMENTS

Our approach is mainly motivated to explore business communication data (namely emails), such as the Enron corpus (Klimt & Yang, 2004). However, due to the lack of ground truth, we will focus our evaluation on research publications and their co-authorship network. Results of dimensionality reduction can be subjective, so as in prior work on dimensionality reduction (McInnes et al., 2018; Sen et al., 2017; Maaten & Hinton, 2008), we will qualitatively compare our approach to a variety of baselines and provide some quantitative experiments.

**Experimental Setup.** Here, we are using the Semantic Scholar[4] Open Corpus (S2) (Ammar et al., 2018) with over 45 million articles covering a range of scientific fields and the AMiner[5] network

---

[3]`https://github.com/redacted/redacted` (link will be published in camera ready version)
[4]`https://api.semanticscholar.org/corpus/`
[5]`https://aminer.org/billboard/aminernetwork`

Table 1: Quantitative evaluation of different landscapes (AM / S2 [/ ENR])

| CLUSTER | DOC2VEC | T-SNE | PCA | MODiR |
|---|---|---|---|---|
| Data Mining | 0.49 / 0.39 | 0.30 / 0.55 | 0.52 / 0.55 | 0.39 / 0.42 |
| Database | 0.49 / 0.82 | 0.64 / 0.34 | 0.47 / 0.34 | 0.69 / 0.32 |
| ML | 0.51 / 0.35 | 0.21 / 0.23 | 0.38 / 0.23 | 0.35 / 0.23 |
| NLP | 0.58 / 0.76 | 0.73 / 0.34 | 0.81 / 0.34 | 0.73 / 0.68 |
| Comp Vision | 0.51 / 0.67 | 0.56 / 0.39 | 0.49 / 0.39 | 0.54 / 0.29 |
| HCI | 0.64 / 0.68 | 0.47 / 0.41 | 0.61 / 0.41 | 0.39 / 0.38 |
| Avg. | 0.54 / 0.61 | 0.49 / 0.37 | **0.54** / 0.38 | 0.53 / **0.39** |
| AtEdge | – | 5.32/4.09/3.89 | 5/3.91/3.6 | **4.79/2.94/2.59** |

(AM) (Tang et al., 2008) published in 2008 with over two million papers by 1.7 million authors. Unlike DBLP however, they not only contain bibliographic metadata, such as authors, date, venue, citations, but also abstracts to most articles, that we use to train document embeddings using the Doc2Vec model in Gensim [6]. Similar to Carvallari et al. (Cavallari et al., 2017), we remove articles with missing information and limit to the six communities Data Mining, Databases, Machine Learning, NLP, Computer Vision, and HCI. This way we discard clearly unrelated computer science articles and biomedical studies for a more fine grained analysis. For in-depth comparisons we use an even smaller subset from S2 of 24 hand-picked authors, their co-authors, and their papers (S2b).

To our knowledge, there are no algorithms that use multiple objectives for dimensionality reduction of high-dimensional data. Popular approaches for traditional dimensionality reduction are tSNE and PCA. As baselines, we use the original optimised implementation of tSNE[7] written in C as provided by the authors. The quantitative evaluation is two-fold: *MODiR* can simulate their behaviour by setting $\theta_3 = 0$ and thus ignoring the objective that incorporates network information.

In our experiments we use the following parameter settings. For tSNE we set the perplexity to $Perp(P_i) = 5$, $\theta = 0.5$ and run it for 1,000 iterations. In *MODiR* we set the neighbourhood size to $k = s = 10$, the negative context size to $l = 20$, and all objective weights $\theta_{\{1,2,3\}} = 1.0$. For a discussion on the influence of hyperparameters, we refer to the supplemental material. The speed of convergence depends on the learning rate $\eta$ and thus dictates the number of maximum iterations. Early stopping with a threshold on the update rate could be implemented. Depending on the size of the dataset and a fixed learning rate of $\eta = 0.01$, *MODiR* generally converges after 10 to 200 iterations, for larger and more connected data it is advisable to use a higher learning rate in the first epoch for initialisation and then reducing it to very small updates. For better comparability, we use a constant number of iterations of $T = 100$.

**Quantitative Evaluation.** As Maaten & Hinton (2008) state, it is by definition impossible to fully represent the structure of intrinsically high-dimensional data, such as a set of document embeddings, in two dimensions. However, stochastic neighbour embeddings are able to capture intrinsic structures well in two dimensional representations (Kobak et al., 2019). To measure this capability, we compare the ability of k-means++ (Arthur & Vassilvitskii, 2007) to cluster the high- and two-dimensional space. We set the number of clusters to the number or research communities ($k = 6$) and calculate the percentage of of papers for each community per cluster. Therefore we assign each community to the cluster with most respective papers and make sure to use a clustering with an even distribution. Results are listed in Table 1 for tSNE, PCA, *MODiR*, and the original high dimensional embedding averaged over five runs. We see, that as expected due to topical overlap of communities, even original embeddings can't be accurately clustered. Interestingly though, there seems to be a significant difference between AM and S2 although the sets of papers intersect, which we assume is due to the fact, that S2 is larger and additionally contains more recent papers. Although PCA

---

[6] https://radimrehurek.com/gensim/; embedding size: 64 dimensions, vocabulary size: 20k tokens, trained for 500 epochs

[7] https://lvdmaaten.github.io/tsne/

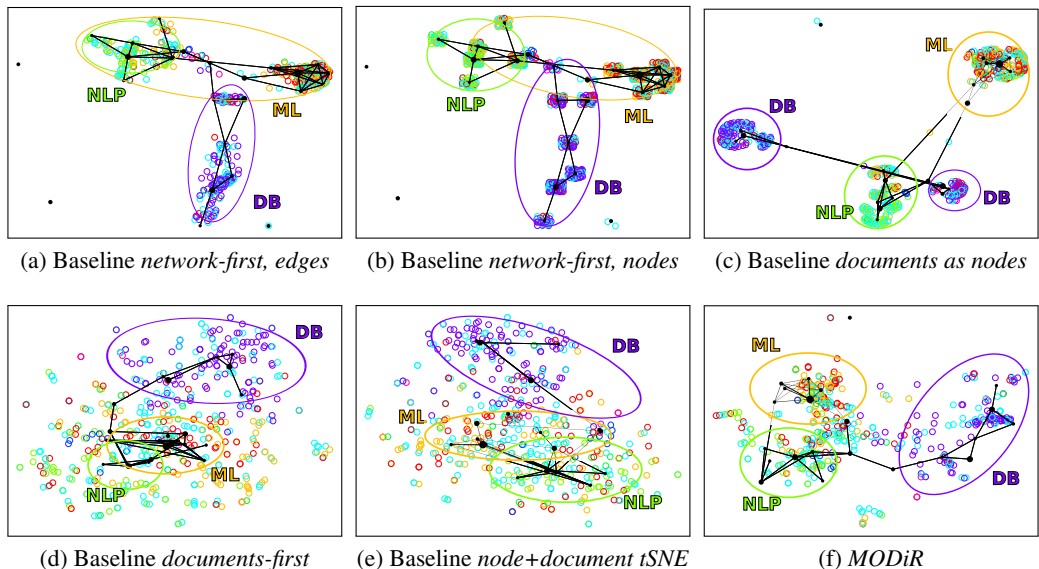

| | | |
|---|---|---|
| (a) Baseline *network-first, edges* | (b) Baseline *network-first, nodes* | (c) Baseline *documents as nodes* |
| (d) Baseline *documents-first* | (e) Baseline *node+document tSNE* | (f) *MODiR* |

Figure 1: Semantic Scholar co-authorship network (S2b), subsampled for readability; (a) the network is laid out first, documents are randomly placed along edges; (b) the network is laid out first, documents are randomly placed around nodes; (c) documents are part of the network layout as nodes in the graph that replace author-author edges; (d) the document landscape is laid out first, nodes are positioned at the centre of their associated documents; (e) tSNE is applied on papers and authors together, where documents are aggregated to represent authors

often does not generate visualisations in which classes can be clearly distinguished, the clustering algorithm is still able to separate them with competitive results compared to tSNE and *MODiR*.

*MODiR* not only aims to produce a good document landscape, but also a good layout of the network layer. Graph layouts are well studied, thus we refer to related work on aesthetics (Purchase, 2002) and readability (Nguyen et al., 2017). While these are very elaborate and consider many aspects, we decided to use Noack's normalised AtEdge-length (Noack, 2007):

$$AtEdge = \frac{\sum_i \sum_j \|\pi_i - \pi_j\|}{|E|} \Big/ \frac{\sum_i \sum_j \|\pi_i - \pi_j\|}{|\mathcal{P}|^2}.$$

It describes how well the space utilisation is by measuring whether edges are as short as possible with respect to the size and density of the graph. Table 1 contains the results. Although the AtEdge metric is comparable for layouts of the same graph, it is not comparable between datasets as can be seen by the fact, that a larger number of edges causes an overall lower score. The AtEdge length produced by PCA is generally better than that of tSNE while *MODiR* outperforms both as our approach specifically includes an optimised network layout. The better performance of PCA over tSNE can be explained by the resulting layouts being more densely clustered in one spot. Although the AtEdge length aims to give a lower score for too close positioning, it is not able to balance that to the many very long edges in the layout produced by tSNE.

**Qualitative Evaluation.** Apart from a purely quantitative evaluation, we use the hand-selected Semantic Scholar dataset (S2b) to visually compare compare network-centric baselines (a-c), document-focused baselines (d-e) and *MODiR* (f) in Figure 1. Papers are depicted as circles where the stroke colour corresponds to the communities, black lines and dots are authors and their co-authorships, size corresponds to the number of publications. For better readability and comparability, the number of drawn points is reduced and three communities are marked.

In Figure 1a we use the weighted co-authorship network drawn using (Fruchterman & Reingold, 1991) and scatter the papers along their respective edges after the graph is laid out. We see, that active collaboration is easy to identify as densely populated edges and research communities of selected areas are mostly coherent and unconnected researchers are spatially separated from others.

Although it is possible to distinguish the different communities in the graph layer,the document landscape isn't as clear. The ML researchers are split apart from the rest of the NLP community, which in turn is overcrowded. Figure 1b uses the same network layout but places articles randomly around their first author, which makes it easy to spot the scientific communities by colour. Lastly, we include papers as nodes and co-authorship edges are connected through them during the network layout in Figure 1c. This produces a very clean looking layout compared with the other baselines, however papers lump together and are not evenly distributed. Furthermore, semantic nuances between papers are mostly lost which becomes most apparent in the now separated database clusters. Also, the semantic overlap between the ML and NLP communities is not noticeable.

Figure 1d positions documents using tSNE and places researchers using Equation 1. We see that articles are positioned on the landscape so that research areas are distinctly recognisable by colour. Papers that could not be assigned to a specific area are scattered across the entire landscape. The collaboration network is laid out surprisingly good. The research interests of the authors are coherent between the network and the document landscape, it even shows the close relation between NLP and ML, while showing a clear separation to database related topics. Nonetheless, the network should be loosened for better readability, for example members of the same research group who frequently co-author papers tend to collide. Unconnected authors are almost not visible as they drift toward densely populated areas in the middle. In Figure 1e, we included authors as virtual documents as the sum of their papers during the tSNE reduction. This shows some improvement, as the network layout is more loose and fewer edges overlap and the issue with collapsing research groups is also mostly mitigated. The semantic overlap of ML and NLP is niceley captured along with the difference to the database papers. However, the network is not clearly readable.

With *MODiR* the three research communities become clearly distinguishable, both in the graph layer and in the document landscape. Nodes of well connected communities are close together, yet are not too close locally, and separate spatially from other communities. The document landscape is laid out more clearly, as papers from different fields are grouped to mostly distinct clusters. Obviously there is still a slight overlap as a result of semantic similarities. As previously pointed out, this visualisation also correctly reveals, that the ML and NLP communities are more closely related to each other (both use machine learning) than to DB. The authorship of documents however can only be conveyed through interaction, so this information is not present in the static visualisations shown here. Based on these results we argue, that the network information improves the (visual) community detection. The document embeddings of articles can only reflect the semantic similarities, which may overlap. In conjunction with information from the co-authorship network, the embeddings are put into their context and thus are more meaningful in a joint visualisation.

## 5 CONCLUSIONS

In this paper we discussed how to jointly visualise text and network data with all its aspects on a single canvas. Therefore we identified three principles that should be balanced by a visualisation algorithm. From those we derived formal objectives that are used by a gradient descend algorithm. We have shown how to use that to generate landscapes which consist of a base-layer, where the embedded unstructured texts are positioned such that their closeness in the *document landscape* reflects semantic similarity. Secondly, the landscape consists of a *graph layer* onto which the inherent network is drawn such that well connected nodes are close to one another. Lastly, both aspects can be balanced so that nodes are close to the documents they are associated with while preserving the graph-induced neighbourhood. We proposed *MODiR*, a novel multi-objective dimensionality reduction algorithm which iteratively optimises the document and network layout to generate insightful visualisations using the objectives mentioned above. In comparison with baseline approaches, this multi-objective approach provided best balanced overall results as measured by various metrics. In particular, we have shown that *MODiR* outperforms state-of-the-art algorithms, such as tSNE. We also implemented an initial prototype for an intuitive and interactive exploration of multiple datasets.

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

# A  APPENDIX

## A.1  ALGORITHM

In Algorithm 1, we formalise our approach for *MODiR* described in Section 3. For the intuition, descriptions, and definitions, we refer to Section 3.

---

**Algorithm 1:** Simplified MODiR Algorithm

---

**Data:** Hypergraph $\mathcal{H} = (\mathcal{V}, \mathcal{E})$ with nodes $\mathcal{V}$ and hyperedges $\mathcal{E}$ as sets of high-dimensional document representations $\mathbb{X}$ and people $\mathbb{P}$; number of iterations $T \in \mathbb{N}$; objective weights $\theta_{\{1,2,3\}} \in \mathbb{R}$; context sizes $k, l, s \in \mathbb{N}$; update size $\eta$

**Result:** document positions $\mathcal{H}^{\mathbb{Y}}$

**begin**
    // initialise landscape (projection matrix $\boldsymbol{W}$)
    $\mathbb{Y} \leftarrow \{\boldsymbol{y}^{(i)} \in \mathbb{R}^2 | \boldsymbol{x}^{(i)} \in \mathbb{X}\}$;
    // pre-processing of contexts
    $\mathbb{X}^k \leftarrow \{(\boldsymbol{x}^{(i)}, \{\boldsymbol{x}^{(j)} | \boldsymbol{x}^{(j)} \in \mathbb{X}^{k,\boldsymbol{x}^{(i)}}\}) | \boldsymbol{x}^{(i)} \in \mathbb{X}\}$;
    $\mathbb{X}^l \leftarrow \{(\boldsymbol{x}^{(i)}, \{\boldsymbol{x}^{(j)} | \boldsymbol{x}^{(j)} \notin \mathbb{X}^{k,\boldsymbol{x}^{(i)}}\}) | \boldsymbol{x}^{(i)} \in \mathbb{X}\}$;
    $\mathbb{Y}^{\mathbb{P}} \leftarrow \{(\boldsymbol{y}^{(i)}, \{\boldsymbol{y}^{(j)} | \boldsymbol{y}^{(j)} \in \mathcal{E}^{\mathbb{Y}}_{\boldsymbol{y}^{(i)}}\}) | \boldsymbol{y}^{(i)}, \boldsymbol{y}^{(j)} \in \mathbb{Y}\}$;
    **for** $t = 1 \ldots T$ **do**
        **foreach** *sampled pairs* $\boldsymbol{x}^{(i)}, \boldsymbol{x}^{(j)}$ *in* $\mathbb{X}^k \cup \mathbb{X}^l \cup \mathbb{Y}^{\mathbb{P}}$ **do**
            $\boldsymbol{y}^{(i)} \leftarrow \psi(\boldsymbol{x}^{(i)})$;
            $\boldsymbol{y}^{(j)} \leftarrow \psi(\boldsymbol{x}^{(j)})$;
            $\Phi \leftarrow \sum \theta_{\{1,2,3\}} \varphi_{\{1,2,3\}}(\boldsymbol{y}^{(i)}, \boldsymbol{y}^{(j)}, \boldsymbol{x}^{(i)}, \boldsymbol{x}^{(j)})$;
            update $W$ by $\eta$ to reduce $\Phi$ with gradient descend;
        **end**
    **end**
    calculate all $\pi$ with Equation 1
**end**

---

## A.2  DETAILS ON DATASETS

The motivation for this paper is to visualise social networks along with their respective text documents, especially email corpora, for exploring and understanding large datasets. We argue, that our approach is applicable to any given dataset with inherent graph structures, so we include a variety of examples for evaluation. We apply *MODiR* to the Enron corpus (Klimt & Yang, 2004) which originally consists of around 600,000 messages belonging to 158 users and Quagga (Repke & Krestel, 2018) to extract individual emails from quoted conversations, remove duplicates, extract additional correspondents from inline metadata, and try to combine the aliases of people. Assessing the quality of a given layout requires very specific domain knowledge including deep understanding of semantic structure across all documents and a close familiarity with entity relations. Email corpora, such as the aforementioned one, lack of gold standards and domain knowledge on our side, so we consider additional sources. Thus we use named entities extracted from business news articles. From the corpus of 448,395 Bloomberg- and 106,519 Reuters news articles (NEW) published by Ding et al (Ding et al., 2014), we select those that contain the search term "commerzbank" as a central entity and consider co-occurrences of organisation entities extracted with AmbiverseNLU (Seyler et al., 2018). This results in a graph where almost all entities are connected to a single central entity that appears in all articles.

Academic co-authorship networks including respective publications have well defined labels provided by venues or communities, so there are no ambiguities or additional annotations. The two processed and publicly available corpora of research articles, the AMiner[8] network (AM) (Tang et al., 2008) published in 2008 with over two million papers by 1.7 million authors and the recently

---
[8]https://aminer.org/billboard/aminernetwork

Table 2: Number of documents, people, and their connections in filtered datasets used in this paper

| DATASET | # DOCUMENTS | # NODES | # EDGES |
|---|---|---|---|
| AMiner (AM) | 49,670 | 56,449 | 110,146 |
| SemanticScholar (S2) | 170,098 | 183,198 | 701,442 |
| SmallScholar (S2b) | 489 | 24 | 39 |
| Enron (ENR) | 189,437 | 32,353 | 950,100 |
| News (NEW) | 3,734 | 2,944 | 5,240 |

Table 3: Number of articles in selected communities from Semantic Scholar (S2) and AMiner (AM)

| LABEL | VENUES | # IN AM / S2 |
|---|---|---|
| Data Mining | KDD, ICDM, CIKM, WSDM | 4,728 / 13,699 |
| Database | SIGMOD, VLDB, ICDE, EDBT | 7,155 / 14,888 |
| ML | NeurIPS, AAAI, ICML, IJCAI | 10,374 / 41,815 |
| NLP | EMNLP, ACL, CoNLL, COLING | 41,815 / 22,523 |
| Comp Vision | CVPR, ICCV, ICIP, SIGGRAPH | 11,898 / 43,558 |
| HCI | CHI, IUI, UIST, CSCW | 8,608 / 33,615 |

published Semantic Scholar[9] Open Corpus (S2) (Ammar et al., 2018) with over 45 million articles. Both corpora cover a range of different scientific fields. Semantic Scholar for example integrates multiple data sources like DBLP and PubMed and mostly covers computer science, neuroscience, and biomedical research. Unlike DBLP however, S2 and AM not only contain bibliographic metadata, such as authors, date, venue, citations, but also abstracts to most articles, that we use to train document embeddings using the Doc2Vec model in Gensim [10]. Similar to Carvallari et al. (Cavallari et al., 2017) remove articles with missing information and limit to six communities that are aggregated by venues as listed in Table 3. This way we reduce the size and also remove clearly unrelated computer science articles and biomedical studies. For in depth comparisons we reduce the S2 dataset to 24 hand-picked authors, their co-authors, and their papers (S2b).

Note, that the characteristics of the networks differ greatly as the ratio between documents, nodes, and edges in Table 2 shows. In an email corpus, a larger number of documents is attributed to fewer nodes and the distribution has a high variance (some people write few emails, some a lot). In the academic corpora on the other hand, the number of documents per author is relatively low and similar throughout. Especially different is the news corpus, that contains one entity that is linked to all other entities and to all documents.

## A.3 Hyperparameter Settings

In Section 4 we gave brief overview of the settings used in the experiments presented above. Here, we provide additional insights from our experiments on different hyperparameter settings for *MODiR*. The context sizes are the most important parameters. The first two objective weights can be ignored, as the context size has similar effects, so we set $\theta_1 = \theta_2 = 1.0$ in all our experiments. Generally, small numbers for $k, l, s$ perform better. This is in line with our expectations, as each item $x(i)$ will also be in the context of its respective neighbours and will therefore amplify its attractive force. A large number for $k$ for example will force all points towards the centre of the canvas or if even larger, produce random scatter as the gradients amplify. In our experiments we use $k = 10$, for datasets with a few thousand samples, $k$ should usually be below $l$. We also found, that the negative context is best with $l = 20$ for all sizes. Furthermore, we set both $\theta_1 = \theta_2 = 1.0$ for all experiments because the influence on selecting $k, l$ is much larger. The graph context is also set to $s = 10$ (in our dataset the number of entities is close to the number of documents), the objective weight can be

---

[9] https://api.semanticscholar.org/corpus/
[10] https://radimrehurek.com/gensim/; embedding size: 64 dimensions, vocabulary size: 20k tokens, trained for 500 epochs

freely adjusted between around $0.8 \leq \theta_3 \leq 1.2$ to set the influence of the entity network. Similar to the semantic neighbourhoods in the first and second objective, the choice of $s$ is significantly more influential than $\theta_3$.

## A.4 LANDSCAPE VISUALISATIONS

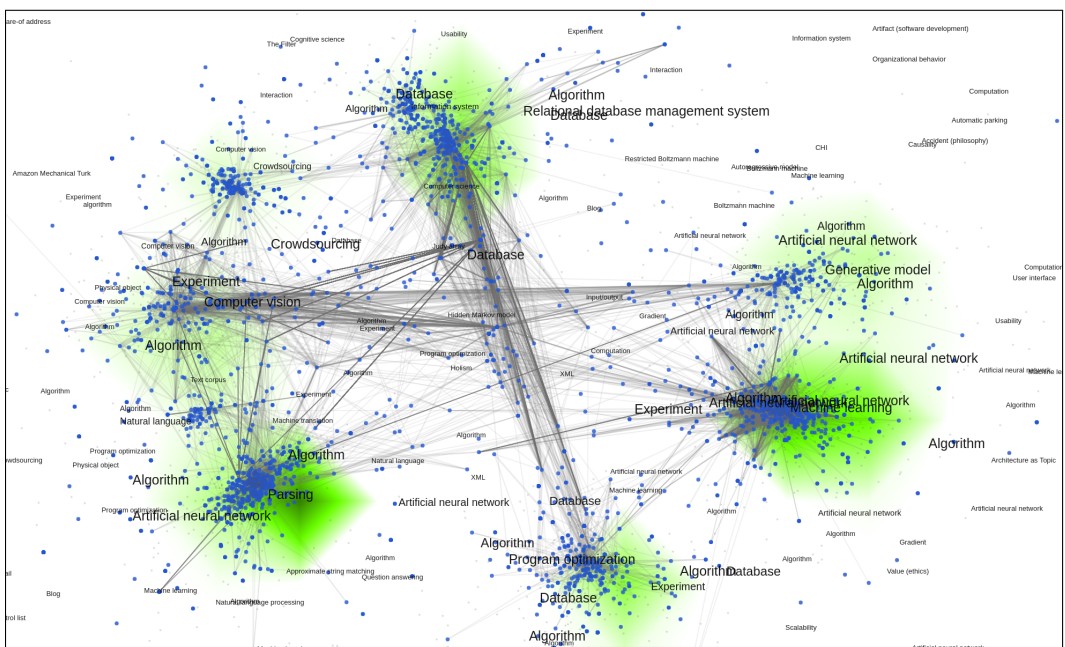

Figure 2: *MODiR* visualisation of Semantic Scholar (S2), all six communities become clear.

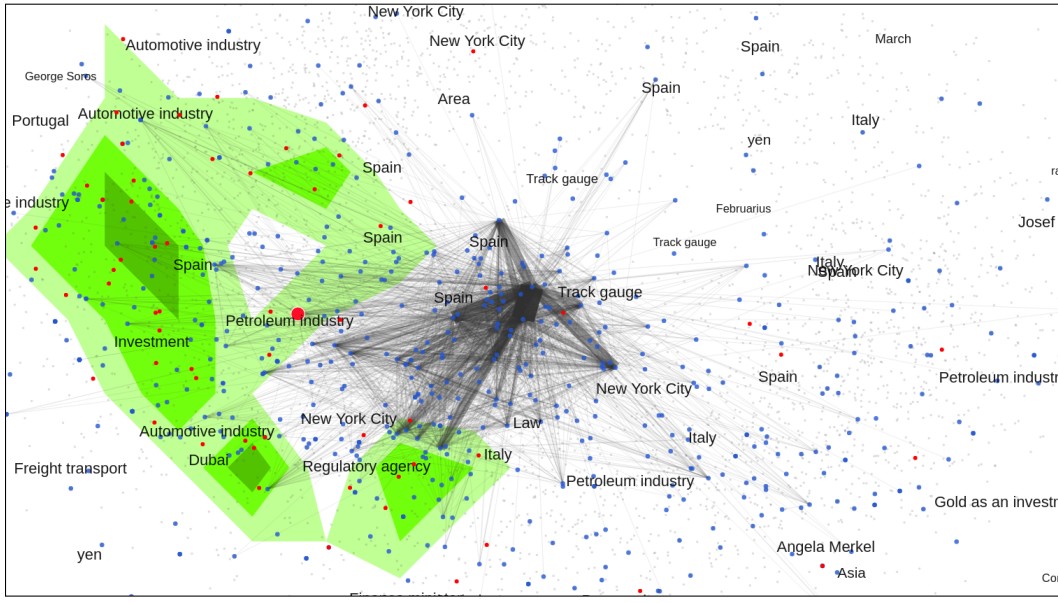

Figure 3: *MODiR* visualisation business news about Commerzbank, highlighted on articles about Volkswagen. All nodes (companies) in the network are connected to the centre node (Commerzbank) and our algorithm still manages to retain the semantic areas.

