# OpenReview forum: "MODiR: Multi-Objective Dimensionality Reduction for Joint Data Visualisation"
_ICLR.cc/2020/Conference — Reject_

### Official Review · AnonReviewer1 · 2019-10-23
**Official Blind Review #1**

**Rating:** 1

**Review:**

Authors introduce a novel algorithm based on multi-objective optimization to jointly position embedded documents and graph nodes in a 2-d landscape.

First, the multi-objective optimization used in this paper is quite confusing. Since the overall error defined in (5) is a weighted summation of three objectives, it is not a real multi-objective optimization where multiple objectives are optimized simultaneously.

Authors define a hypergraph to jointly model documents and entities. However, there lacks the novelty on the objectives defined in this paper. The optimization using gradient descent is also trivial.

It is hard to understand why equations (2) and (3) share the same objectives but the two sets of samples have totally different meanings. Moreover, both objectives are summed uniformly as shown in the experimental setting. Please explain it in details.

For the quantitative evaluation, Table 1 shows the percentage of papers for each community per cluster. There are many numbers for a single clustering problem. It might be preferred to use a single measurement such as normalized mutual information or accuracy for clustering evaluation. In this way, it is easy to judge the compared methods.


**Experience Assessment:**

I have published one or two papers in this area.

**Review Assessment: Checking Correctness Of Derivations And Theory:**

I assessed the sensibility of the derivations and theory.

**Review Assessment: Checking Correctness Of Experiments:**

I assessed the sensibility of the experiments.

**Review Assessment: Thoroughness In Paper Reading:**

I read the paper thoroughly.

---

### Official Review · AnonReviewer3 · 2019-10-23
**Official Blind Review #3**

**Rating:** 3

**Review:**

[Contribution summary]
Authors propose MODiR, a dimensionality reduction approach that utilizes both semantic and structural features. Authors report quantitative (metric: precision of k-means++ clustering & AtEdge-length) and qualitative analysis (visualization).

[Comments]
Experimental analysis could improve e.g. by including more baselines (e.g. other baselines the authors cite in the paper), or by providing more quantitative analysis - e.g. performance at varying degree of completeness / artificial perturbation of the graph. In the current manuscript, the quantitative analysis does not immediately show the benefit or the characteristics of the proposed approaches. In the main section of the manuscript, the analysis is limited to citation networks.

While the idea of leveraging both semantic and structural information for the purpose of dimensionality reduction seems to be new and interesting, similar approaches have been explored for text classification tasks (e.g. via graph convolutional networks).

**Experience Assessment:**

I have read many papers in this area.

**Review Assessment: Checking Correctness Of Derivations And Theory:**

I carefully checked the derivations and theory.

**Review Assessment: Checking Correctness Of Experiments:**

I carefully checked the experiments.

**Review Assessment: Thoroughness In Paper Reading:**

I read the paper thoroughly.

---

### Public Comment · ~Tim_Repke1 · 2022-01-04
**Published at JCDL and IUI 2020**

A revised version of this work has been published at JCDL 2020, including a demo at IUI 2020.
If you would like more information about the project, please visit https://hpi.de/naumann/s/modir

> Tim Repke, Ralf Krestel: Visualising Large Document Collections by Jointly Modeling Text and Network Structure. Proceedings of the Joint Conference on Digital Libraries (JCDL), 2020
> Tim Repke, Ralf Krestel: Exploration Interface for Jointly Visualised Text and Graph Data. Proceedings of the International Conference on Intelligent User Interfaces Companion (IUI), 2020

---

### Decision · Program_Chairs · 2019-12-19

**Decision:**

Reject

**Comment:**

There is a consensus among reviewers that the paper should not be accepted. No rebuttal was provided, so the paper is rejected.